# Defatted Seed Residue of *Cucumis Melo* as a Novel, Renewable and Green Biosorbent for Removal of Selected Heavy Metals from Wastewater: Kinetic and Isothermal Study

**DOI:** 10.3390/molecules27196671

**Published:** 2022-10-07

**Authors:** Taslim Akhtar, Fozia Batool, Sajjad Ahmad, Eida S. Al-Farraj, Ali Irfan, Shahid Iqbal, Sami Ullah, Magdi E. A. Zaki

**Affiliations:** 1Government Associate College for Women, Mandi Bahauddin 50400, Pakistan; 2Department of Chemistry, University of Sargodha, Sargodha 40100, Pakistan; 3Department of Chemistry, UET-Faisalabad Campus, Faisalabad 37630, Pakistan; 4Department of Chemistry, College of Science, Imam Mohammad Ibn Saud Islamic University (IMSIU), Riyadh 11623, Saudi Arabia; 5Department of Chemistry, Government College University Faisalabad, Faisalabad 38000, Pakistan; 6Department of Chemistry, Jauharabad Campus, University of Education Lahore, Lahore 41200, Pakistan; 7Department of Chemistry, Sargodha Campus, University of Lahore, Sargodha 40100, Pakistan

**Keywords:** biosorption, heavy metals, Taguchi method, kinetic studies, isotherms, defatted seed residue, *Cucumis melo*, waste biomass valorization

## Abstract

The present work was aimed at studying the biosorption of two important heavy metals, viz. Pb and Cr, using defatted seed residue of *Cucumis melo* as biosorbent. As this study for the biosorption of the selected biosorbent is being carried out for the first time, optimization of the% sorption was carried out with the help of Taguchi method. Three most influential experimental factors were taken into account for this purpose, including the amount of sorbent, amount of sorbate and shaking time. For Pb, maximum% sorption was found to be 94.1%, using 2 g of sorbent and 5 ppm of sorbate after 2 h of shaking. Similarly, for Cr, maximum% sorption was 92.5% using 2 g of sorbent, 10 ppm of sorbate and 3 h of shaking. For Pb, the highest% contribution, which was determined by ANOVA, was given by the amount of sorbate (54.7%) followed by the amount of sorbent (38.8%) and the least contribution was given by the shaking time (6.47%). Similarly, for Cr, the highest% contribution, which was determined by ANOVA, was given by the amount of sorbate (75%) followed by the amount of sorbent (16%) and the least contribution was given by the shaking time (8.65%). Kinetic and isothermal studies were also performed to understand the nature of adsorption mechanism. For this purpose, linear and non-linear forms of three sorption isotherms were employed including Freundlich, Langmuir and Dubnin–Radushkevich isotherm. From these observations, it can be concluded that the defatted seed residue of *Cucumis melo* can be regarded as a novel, renewable, green and cost-effective biosorbent for removal of heavy metals from wastewater.

## 1. Introduction

Rapid industrialization has led to tremendous disposal of heavy metals into the environment. This is due to the rapid expansion of industries such as metal plating facilities, fertilizer industries, mining operation, batteries, tanneries, paper industry and pesticides discharging heavy wastewater into the environment. These industrial and agricultural wastes contaminate water with heavy metals which have drastic effects on living organisms through the food chain [1,2,3]. These metals buildup and cause different drastic effects for human and amphibian life. Notably, these metals cause harm to the reproductive system, liver, kidneys, enzyme inhibitors and so forth [4].

The toxicity of heavy metals to human health and the environment has been an interesting subject to biologists for many years. Many conventional methods used to remove heavy metal ions from waste resources include chemical oxidation and reduction, electrodialysis, evaporation, electro precipitation, liquid extraction and ultrafiltration, etc., from dilute solutions [5]. All these conventional methods are very expensive and time-consuming. These methods also do not completely remove metals from water. For high strength and low volume of wastewater, the best method for removal of heavy metals is biosorption. Biological methods such as bioaccumulation/biosorption may provide an alternative to physio-chemical methods [6].

The biosorption method suggests the minimization of chemical and biological sludge. This method is technically easy, cost-effective and time-saving. For this process, naturally present biosorbents are preferred due to their large surface area and greater capacity of adsorption for heavy and toxic metals. These biosorbents are the most widely consumed sorbents in the world. At present, some biosorbents are gaining much attention by researchers due to their greater heavy metal adsorbtion capacity [7,8]. The most studied biosorbents are agricultural waste, such as spent coffee grounds, organum stalks, walnut shells, tea waste, corncob, banana and orange peels, etc. The benefit of using these adsorbents is the presence of functional groups of organic compounds (such as polysaccharides, alkaloids, etc.) that are able to interchange heavy metal ions by chemical reaction [9,10,11].

In this paper, we have explored for the first time the adsorption of two important heavy metals, viz. Pb and Cr, using the defatted seed residue of *Cucumis melo.* This defatted seed residue is actually a by-product of oil extraction from the seed of *C. melo* and is usually discarded. This oil is recently gaining importance due to its consumption as an edible oil as well as a biodiesel feedstock [12,13]. *C. melo* is a species of the family *Cucurbitaceae* and many of its varieties are grown in Pakistan (generally known as melons), mainly for utilization as fresh fruits [12]. The seeds of these fruits are usually discarded and can be availed in bulk quantity as an agro-waste material. Overall, various species of this family, specifically *Citrullus* and *Cucumis* (generally called melons), are grown as a main food crop in different tropical and sub-tropical countries of the world [13]. Since the said biosorbent is being experimented on for the first time, an optimization study has also been carried out.

## 2. Results and Discussion

### 2.1. Design of Experiment Utilizing L9 Orthogonal Array

At present, the Taguchi method is considered as the standard version of DoE. An important feature of this method is that it does not involve all the possible combinations of parameters, but only uses a few standard parameters. It employs the L9 orthogonal array to estimate the standard conditions for the process under study, i.e., the biosorption of heavy metals.

### 2.2. Estimation of the Optimum Conditions for the % Sorption by Taguchi Method

Table 1 and Table 2 show the detailed experimental matrix, as well as results of these experiments for the mean % sorption, along with the corresponding SNR values for Pb and Cr, respectively, while using the defatted seed residue of *C. melo* as a novel, renewable and green biosorbent. Since the objective of the present optimization study was to find out the experimental conditions that offer maximum % sorption, this was accomplished using the LTB model of SNR. As is clear from Table 1, experiment #4 shows the maximum % sorption, up to 94.1%, with a corresponding SNR value of 39.78 for Pb. Similarly, Table 2 shows that experiment #5 offers the maximum % sorption, i.e., 92.5% with an SNR value of 39.32, for Cr. These values of % sorption have been found comparable with those reported earlier using other agricultural wastes used for the same purpose, especially defatted *Carica papaya* seeds [9,10,11].

The three most influential parameters that are independent of the type and amount of wastewater to be decontaminated from these heavy metals were the focus of this study, suggesting that the biosorption process be carried out at pH = 7 (i.e., in neutral conditions) and preferably at room temperature, as this would make the sorption process cost-effective as well as green (in terms of avoiding the requirement to maintain specific temperature and pH). This was also considered appreciable, as quite good sorption efficiency (94.1% for Pb and 92.5% for Cr) was obtained.

Melon seeds have been reported to contain mainly protein, fat and fiber—all in crude form—up to 25%, 25% and 23.3%, respectively. The next major component, carbohydrates, is present up to 19.8% in these seeds [13]. It can be suggested that the functional groups present in these seed components are responsible for the high sorption efficiency observed in this study, especially the carboxylic groups of carbohydrates and proteins, as well as the hydroxyl groups of carbohydrates.

The suggested biosorbent is produced as a result of oil extraction from seeds of *C. melo* and hasno further use and must be simply discarded. As a by-product of the oil extraction, it has been termed here as defatted(de-oiled) seed residue of *C. melo,* and will pose no cost at all if used as such. However, if the biosorbent is prepared for the purpose of biosorption only, the seeds are available in the form of domestic as well commercial waste material after consumption of the melon as fresh fruits and can be collected and processed with a minimum cost as well as minimal human labor. In this case the extracted oil will be a valuable by-product.

The values of SNR_L_ (level mean SNRs) for three experimental factors at their specificlevels are shown in Table 3 for Pb and Cr. In this table, the SNR_L_value for parameter A at its 1st level was calculated using the values of SNR for experiment #1, 2 and 3, i.e., the SNR values of those experimentswhich were carried out using level 1 of this parameter.

Similarly, for its SNR_L_ value at the 2nd level, the SNR values of those experiments were used where its level was kept at 2, which includes experiment #4, 5 and 6, and so on the SNR_L_ value at level 3. The SNR_L_ values of each parameter at three different levels, in fact, point towards its effect on the% sorption of the metals under study. The higher the value of SNR_L_ of a parameter at a particular level, the higher the effect on % sorption at that level [12,14].

The SNR_L_ values of parameters A, B and C at levels 2, 2 and 3, respectively, were found to be maximum for Pb, which show that these are the optimal levels for the % sorption of Pb on the biosorbent under study, i.e., defatted melon seed residue. These constitute 2 g amount of sorbent, 10 ppm of sorbate and 3 h of shaking time, respectively.

The same calculation was carried out for Cr. The optimal levels of three parameters according to values of SNR_L_ include levels 1, 2 and 3 for parameters A, B and C, respectively. These correspond to 1 g of sorbent, 10 ppm of sorbate and 3 h of shaking time.

### 2.3. Analysis of Variance

With the help of SNR analysis, although it is possible to attain the optimum level of each parameter, as well as an optimum set of parameters that provide maximum yield of the preferred product, it is not yet recognized how much each parameter has added to the output and which parameter has contributed more significantly.

However, this can be recognized by carrying out a statistical analysis of variance (ANOVA) of the response data. For this reason, computation of sum of squares is necessary. Table 3 represent the computed sum of squares (SS) and % contribution of each process parameter for the sorption of Pb and Cr, respectively.

The results of sum of squares (SS) and % contribution can be used to determine the most significant parameter, as the parameter with highest % contribution will influence % sorption the most [15].

The % contribution of the parameters towards maximum output of the adsorption process on the same adsorbent, i.e., defatted seed residue of *C. melo,* is different for the two metals under study. Table 4 represents the % contribution of the parameters A, B and C for adsorption of Pb that include maximum share from amount of sorbent up to 54.7%, followed by the amount of sorbent that was 38.8%, while the shaking time contributed up to 6.47%.

Similarly, the results of sum of squares (SS) and % contribution of the three parameters for Cr are also shown in Table 4. For Cr, maximum contribution is received from the amount of sorbate, up to 75%, then by the shaking time up to 16% and least of all by the amount of sorbent, up to 8.65%.

The optimal levels of the parameters were employed to sort out the % sorption for both the metals in triplicate experiments and the mean % sorption was found to be 94% and 92%, for Pb and Cr, respectively, which is quite close to that obtained in experiment #4 and 5 for the respective metals.

### 2.4. Equilibrium Isotherms

Regarding the removal of effluents from a system, a particular design needs to be optimized in order to sort out an appropriate correlation for experimental data. This is known as equilibrium isotherm or adsorption isotherm. These isotherms represent the amount of sorbate that is adsorbed by unit weight of the adsorbent and are, therefore, exploited to design the adsorption system. An equilibrium concentration of sorbate at constant temperature is utilized for constructing these isotherms.

Many isotherms have been proposed by researchers such as Langmuir, Freundlich, Elovich, Dubnin–Radushkevich, Temkin and Redlich–Peterson [16,17].

In the present work, adsorption isotherms were developed by employing both the linear and nonlinear models of Freundlich, Langmuir and Dubnin–Radushkevich, while varying the initial concentration of the sorbate from 5–25 ppm, for both the metals under study, i.e., Pb and Cr.

#### 2.4.1. Freundlich Isotherm

The Freundlich isotherm was basically developed for heterogeneous systems and represents the concept of multilayer adsorption on the surface of the sorbent [18]. Table 5 shows the parameters calculated for both its linear and nonlinear forms. K_F_, known as the Freundlich adsorption capacity, predicts whether a system is suitable for adsorption or not. A value of K_F_ between 1 and 20 indicates that the adsorption can be considerable. In the present work, the value of K_F_ for Pb was found to be 7.39 for both linear and nonlinear approaches, while for Cr it was found to be 2.079 in both models of the Freundlich adsorption isotherm.

Similarly, the value of n, the adsorption intensity, shows whether or not a sorbent is fit (if its value is greater than 1) for adsorption to occur. A significant value of R^2^ (0.9111 for both metals) indicates that the model is fairly suitable for adsorption of both metals on the selected biosorbent, i.e., defatted seed residue of *C. melo*.

#### 2.4.2. Langmuir Isotherm

This is based on monolayer adsorption of sorbate (e.g., metal ion) on the surface of the sorbent, considering the energy of the adsorption system constant.

R_L_ is obtained using values of initial concentration and b. If its value is from 0–1, the system under study is considered good enough for the adsorption process to take place and the model is said to be fit for this process. Table 5 shows that the results for the present study are in this range. Similarly, a close correlation was obtained between the predicted and experimental results with a low value of RSS (residual sum of squares) for Cr in both linear and non-linear approaches, making the model fit for it. However, for Pb, the value of RSS was low for the linear model but high for the non-linear approach.

#### 2.4.3. Dubnin–Radushkevich Isotherm

This isotherm was basically designed as an empirical model for the sorption of vapors on the surface of solids. However, it has been applied successfully on other heterogeneous systems as well, such as solids and liquids. This approach is thought of as rather more general than Langmuir’s model, since a homogeneous surface and a constant adsorption potential are not assumed in its derivation.

In the present work, the value of ɛ calculated for Pb and Cr was 0.37 and 0.42, respectively. Therefore, it suggests the physical nature of adsorption on the selected biosorbent, i.e., defatted seed residue of *C. melo*.

### 2.5. Adsorption Kinetics

Adsorption kinetics have gained basic importance in determining the rate of solute uptake/sorption and the time taken for the adsorption process to occur [19,20].

For the present work, the kinetic study was carried out for both Pb and Cr at different time intervals, using both linear and non-linear forms of pseudo-first order and pseudo-second order kinetics.

#### 2.5.1. Pseudo-First Order Kinetics

In order to study the pseudo-first order kinetics, log(Q_e_*−Q_t_*) was plotted against t, the time interval in minutes, and the values of k and Q_e_ were obtained from the slope of the line and intercept, respectively. The following equation was employed to find out the initial sorption rate, h [21].
$$H=k_{2}Q_{e}^{2}$$

Since a significant value of R^2^ was obtained for the linear form of the model (0.9897 for Cr and 0.8965 for Pb), it can be assumed that the adsorption of both the metals on to the defatted seed residue of *C. melo* follows pseudo-first order kinetics. MS-Excel 2010 was employed to obtain the non-linear form of pseudo-first order kinetics [22].

#### 2.5.2. Pseudo-Second Order Kinetics

Pseudo-second order kinetics are basically applied for determining the initial adsorption rate when there are low initial concentrations of the sorbent [22,23]. Different forms of pseudo-second order kinetics, both linear and non-linear, are given in Appendix A for P band Cr, and Appendix A show the results of these models as applied on experimental data. R^2^, the coefficient of determination, was found to be quite high for type 1 for both Pb as well as Cr, predicting the best fitting of this model on the adsorption data of these metals.

## 3. Materials and Methods

### 3.1. Chemicals and Reagents

The chemicals and reagents required to carry out this work include lead nitrate Pb(NO_3_)_2_, chromium sulphate Cr_2_(SO_4_)_3_and n-hexane. These chemicals were bought from Sigma Aldrich (Burlington, MA, USA). To prepare all solutions, deionized water was utilized. Refined nitrogen gas (99%) was acquired from Mega Mount Industrial Gases Sdn. Nitrogen gas was utilized in atomic absorption spectrometry for the assurance of metal fixation after the biosorption process.

### 3.2. Procurement of Raw Material

Melons were obtained from rural areas of Sargodha. The seeds were separated from the fruit manually, washed with distilled water, dried in sunlight for a few days and stored in airtight jars. The seeds were ground using an electric grinder before being subjected to oil extraction.

### 3.3. Preparation of Adsorbent and Sorbate Solutions

For the preparation of the adsorbent, i.e., defatted seed residue of melon, extraction of oil from melon (*C. melo*) seeds was accomplished using a Soxhlet assembly and heating mantle. The dried seeds were ground using a domestic electric grinder. An amount of 30 g of melon seed powder was transferred to a Soxhlet extractor fitted with a 1 L round-bottom flask using n-hexane as extraction solvent. The extraction of seed oil was completed in approximately 6 h using a heating mantle set at 50 °C. Then, defatted seed residue was removed from the extractor and dried in open air. After this, defatted seed residue was stored in polythene bags for further use as adsorbent.

Stock solutions (1000 ppm for both Pb and Cr) were prepared from lead nitrate and chromium sulphate, respectively, using distilled water. Working solutions were prepared and adsorbent was applied indifferent amounts according to the experimental design prepared using Taguchi method employing orthogonal array. Three volumetric flasks (250 mL) were taken to prepare working solution (5 ppm, 10 ppm, 15 ppm) of Pb as well as Cr. This solution was transferred to the conical flasks with specified amount of the adsorbent (defatted seed residue of *C. melo*) in different experiments as detailed in next section. These flasks were subjected to orbital shaker for 1 h at 100 rpm shaking speed.

### 3.4. Metal Determination by Atomic Absorption Spectrophotometer

After shaking process for 1 h, metal ion solutions were filtered to remove the adsorbent and the concentration of the metal ions, both before as well as after the process of adsorption was analyzed by using atomic adsorption spectrometric technique (AAS).

The % sorption was calculated using the following relation:(1)% sorption=CfCi

*C_f_* = Metal ion concentration after the adsorption process

*C_i_* = Metal ion concentration before the adsorption process

For this purpose, calibration curve method was employed. Each experiment was repeated thrice to obtain mean % sorption.

### 3.5. Selection of Parameters for the Optimized % Sorption by Selected Biosorbent

There are different parameters that influence the mean % sorption; however, the three most influential parameters were selected for the optimization of % sorption by the selected biosorbent. These include amount of sorbent, amount of sorbate, and contact/shaking time. Three levels of selected parameters were selected for the sake of optimization of % sorption. These maximum and minimum levels of each parameter were decided after analyzing the effect of each parameter on % sorption of the metal ion under study. The three levels of each selected parameter are shown below in Table 6.

### 3.6. Design of Experiments Utilizing Taguchi Method/Orthogonal Array

Taguchi method is one of the standardized versions of the design of experiments (DoE). It exploits the L9 orthogonal array to design the experimental matrix [12,14]. This statistical tool works on the basis of number of parameters and their variation level of each parameter separately. Number of levels (L) and chosen control parameters (P) are helpful to decide the least number of experiments (N). This can be demonstrated as following relationship:N = (L − 1) P + 1(2)

Three parameters at three levels resulted in 9 different combinations as shown in Table 7.

### 3.7. Signal to Noise Ratio (SNR) and Analysis of Variance (ANOVA)

In Taguchi method, use of loss function has been suggested in order to calculate the deviance between the experimental and desired value of performance features. This value of loss function is further converted into signal to noise ratio (SNR). SNR is then employed to compute the extent of deviation by the quality function from the expected value. Depending upon objectives of the problem, we use SNR larger the better (LTB) for minimization of problem. SNR value for LTB models can be calculated as given below:(3)$$Larger\;the\;better-SNRi=-10\;log\;\left(1/n\right)\left[\;\overset{n}{\underset{j=1}{\sum }}1/y_{j}^{2}\right]$$
where *n* is total number of *SNR* levels that are contributed to determine the influence of parameter A on the biosorption process. y_j_ in this formula is mean value of response.

For calculating the percentage contribution of each factor, following equations are employed:(4)% contribution of a factor =SSfSST×100 where SS_f_ stands for the sum of squares for f^th^ factor, while SS_T_ represents the total sum of squares of the entire parameters [12,14,15].

### 3.8. Equilibrium Isotherms

In order to follow the adsorption pathway and equilibrium relationship between sorbent and the sorbate under study, it is imperative to design appropriate adsorption/equilibrium isotherms. These isotherms predict the proper parameters, as well as the behavior of the sorbent towards various sorption systems [21]. For this purpose, both linear and non-linear models are employed using MS-Excel^®^ 2007.

In order to obtain Freundlich isotherm, log C_ad_ was plotted against log C_e_ while n and K_F_ are the constants found as slope and intercept, respectively.

For calculations of Langmuir model, the initial concentration was varied from 5–25 ppm, using 1 g of adsorbent for 1 h shaking. Plotting C_e_/C_ad_ versus C_e_ furnished the Langmuir adsorption isotherm. Significant R^2^ value showed that the model was fit for the adsorption process under study. The parameters calculated in this approach include Q_o_ (the mass of metal ion per unit mass of the sorbent in mg/g), b (Langmuir constant) and R_L_ (a dimensionless constant).

Dubnin–Radushkevich isotherm has shown very satisfactory prediction to sort out the nature of adsorption, i.e., whether it is physical or chemical. This requires calculation of E according to the following relationship:(5)$$E=\frac{1}{\sqrt{{2k}_{\mathrm{ad}}}}$$

k_ad_ was obtained as slope of the plot between l_n_C_ad_ and ε^2^, i.e., Polanyi potential as given by the following equation:(6)$$\varepsilon =\mathrm{RTln}\left(1+\frac{1}{C_{e}}\right)$$

This relationship shows that ε is based on natural gas constant, temperature and equilibrium concentration of the sorbate. The value of ε below 8 kJ/mol shows physical nature of adsorption while between 8–16 kJ/mol is accepted to be chemical adsorption [18].

### 3.9. Adsorption Kinetics

The information provided by the kinetic studies, regarding the batch adsorption process, includes optimum conditions, mechanism of sorption, as well as possible rate controlling step. This can be accomplished by applying linear and non-linear forms of pseudo-first order and pseudo-second order kinetics on the adsorption data [18].

To find out the effect of contact time (10–60 min) on adsorption, an initial concentration of 100 ppm for both Pb and Cr was prepared separately, and its 100 mL was used for subsequent study. An amount of 0.5 g of the sorbent, i.e., defatted seed residue of *C. melo,* was added to the sample of each metal and samples were subjected to shaking at 160 rpm.

Following the fixed time intervals, the sample of each metal was removed from the flask for determining the concentration of the metal through atomic absorption spectroscopy. The relationship given below was used to calculate the amount of metal adsorbed at the fixed time interval:(7)$$Q_{t}=\frac{Q_{o}\;-\;Q_{e}}{W_{adsorbent}}$$
where *Q_t_*, Q_o_ and Q_e_ are the amounts of metal adsorbed at any time ‘t’, initial concentration and equilibrium concentration, respectively. Similarly, V and W_adsorbent_ represent the volume of metal solutions taken and the amount of adsorbent in grams.

Pseudo-first order Kinetics

The following equations were used to calculate pseudo-first order kinetics for the adsorption system under study:(8)$${\ln(Q}_{e}-{\;Q}_{t}{)=\ln(Q}_{e})\;-{\;k}_{1}t\;(\mathrm{linear}\;\mathrm{form})$$
(9)$$Q_{t}{=Q}_{e}(1-{\;e}^{-k_{1}t})\;(\mathrm{non}-\mathrm{linear}\;\mathrm{form})$$

Pseudo-Second Order Kinetics

Both linear and non-linear form were applied for evaluating pseudo-second order kinetics on adsorption system, as below [18]:(10)$$\frac{t}{Q_{t}}=\frac{1}{k_{2}Q_{e}{}^{2}}+(\frac{1}{Q_{e}})\;t\;(\mathrm{linear}\;\mathrm{form})$$
(11)$$Q_{t}=\frac{k_{2}Q_{e}^{2}t}{{1+k}_{2}Q_{e}t\;}(\mathrm{non}-\mathrm{linear}\;\mathrm{form})$$

## 4. Conclusions

Wastewater treatment regarding the removal of toxic heavy metals has presently gained enormous interest due to the detrimental effects of heavy metals on humans and other living organisms. Chromium and lead are among the most familiar wastewater contaminants owing to their highly toxic effects even if present in a very low concentration. The present work represents the study carried out on biosorption of these heavy metals (Pb and Cr) by using defatted seed residue of *C. melo* as a novel, cost-effective and environmentally benign biosorbent. Optimization of the % sorption of the selected heavy metals on this adsorbent was also carried out using the Taguchi method, as the reported biosorbent has been explored for the first time. High % sorption for both the metals was obtained, viz. 94% for Pb and 92% for Cr, which proves the newly explored biosorbent to be an efficient one. Kinetic and isothermal studies were also performed to check the adsorption mechanism. It was found that these models were quite fit for the adsorption system under study. The Dubnin–Radushkevich isotherm suggested that the adsorption was physical in nature. Keeping in mind that this biosorbent is an agricultural wasteproduct that is biodegradable, it can be attributed to be an eco-friendly, economical and green biosorbent for heavy metals [23]. In addition to a contribution towards wastewater treatment, the present work can be regarded as an effort for waste biomass valorization as well.

## Figures and Tables

**Table 1 molecules-27-06671-t001:** Mean percentage sorption and SNRs for the experiments designed by L9 orthogonal array for Pb on defatted seed residue of *C. melo.*

Exp No.	A	B	C	% Sorption			Mean% Sorption	SNRs
				Trial 1	Trial 2	Trial 3		
1	1 g	5 ppm	1 h	90	91	91.2	90.77	39.172
2	1 g	10 ppm	2 h	86	85	85.6	85.5	38.66
3	1 g	15 ppm	3 h	86	86.6	88	86.9	38.79
4	2 g	5 ppm	2 h	94	94.1	94.3	94.1	39.78
5	2 g	10 ppm	3 h	92	92.4	93	92.4	39.31
6	2 g	15 ppm	1 h	84	83.2	83	83.4	38.44
7	3 g	5 ppm	3 h	86	85.6	87	86.5	38.86
8	3 g	10 ppm	1 h	82	83.2	84.6	83.2	38.42
9	3 g	15 ppm	2 h	82	82	83	82.33	38.32
SNR_T_ = 38.86								

A = Amount of sorbent, B = amount of sorbate, C = time, SNR = signal to noise ratio.

**Table 2 molecules-27-06671-t002:** Mean percentage sorption and SNRs for the experiments designed by L9 orthogonal array for Cr on defatted seed residue of *C. melo*.

Exp#	A	B	C	% Sorption			Mean% Sorption	SNRs
				Trial 1	Trial 2	Trial 3		
1	1 g	5 ppm	1 h	92	91	89	90.66	39.14
2	1 g	10 ppm	2 h	87	88	88	87.66	38.86
3	1 g	15 ppm	3 h	86	85	85.7	85.6	38.66
4	2 g	5 ppm	2 h	85	84.5	84	84.5	38.53
5	2 g	10 ppm	3 h	92	92.5	93	92.5	39.32
6	2 g	15 ppm	1 h	83.6	84	82.2	84.2	38.42
7	3 g	5 ppm	3 h	86	87	87.3	86.77	38.75
8	3 g	10 ppm	1 h	85	84.5	85	85	38.57
9	3 g	15 ppm	2 h	82	83.3	82.1	82.1	38.33
SNRT= 38.78								

A = Amount of sorbent, B = amount of sorbate, C = time, SNR = signal to noise ratio

**Table 3 molecules-27-06671-t003:** Level mean SNR (SNR_L_) for different parameters levels for Pb and Cr.

Parameter		SNR_L_ for Pb			SNR_L_ for Cr		
		1	2	3	1	2	3
**A**	**Amount of sorbent**	38.89	39.17	38.53	38.88	38.76	38.55
**B**	**Amount of sorbate**	39.28	38.8	38.51	38.81	38.91	38.47
**C**	**Time**	38.68	38.92	38.98	38.71	38.83	38.91

**Table 4 molecules-27-06671-t004:** Percentage contribution of selected process variables towards % sorption of Pb and Cr on defatted seed residue of *C. melo.*

Parameters	Pb		Cr	
	SS_f_	% Contribution	SS_f_	% Contribution
**Amount of sorbent**	0.4259	54.7%	0.0131	8.65%
**Amount of sorbate**	0.3025	38.8%	0.1139	75%
**Time**	0.0504	6.47%	0.0243	16%

**Table 5 molecules-27-06671-t005:** Linear and non-linear parameters of isothermal models for biosorption of Pb and Cr on defatted seed residue of *C. melo.*

Parameters	Linear Method		Non-Linear Method	
	Pb	Cr	Pb	Cr
**Freundlich Isotherm**				
K_F_ (mg/g) (L/mg) ^n^	7.392	2.0792	0.0499	0.0499
n	1.103	1.3869	1.84	1.8439
R^2^	0.6543	0.9944	0.9616	0.9616
**Langmuir Isotherm**				
Qo (mg/g)	8.591	38.61	8.591	38.61
b (L/mg)	0.2566	0.1664	0.2566	0.1664
R_L_				
R^2^	0.4607	0.9938		
**Dubinin–Radushkevich Isotherm**				
Kad (mol^2^/kJ^2^)	3.6218	2.7691	3.6218	2.7691
q_s_(mg/g)	1.6007	1.3801	1.601	1.3801
R^2^	0.6779	0.9043		

**Table 6 molecules-27-06671-t006:** Parameters selected for biosorption of Pb and Cr and their levels.

Parameters		Levels		
		1	2	3
**A**	**Amount of sorbent (g)**	1	2	3
**B**	**Amount of sorbate (ppm)**	5	10	15
**C**	**Time (h)**	1	2	3

**Table 7 molecules-27-06671-t007:** Experimental design for biosorption of Pb as well as Cr on defatted seed residue of *C. melo* (by L9 orthogonal array).

Experiment No.	A (Amount of Sorbent)	B (Amount of Sorbate)	C (Time)
1	1	1	1
2	1	2	2
3	1	3	3
4	2	1	2
5	2	2	3
6	2	3	1
7	3	1	3
8	3	2	1
9	3	3	2

## Data Availability

The data are available in the Appendix A section.

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
