# Peer review of "Defatted Seed Residue of Cucumis Melo as a Novel, Renewable and Green Biosorbent for Removal of Selected Heavy Metals from Wastewater: Kinetic and Isothermal Study"

_molecules, 2022, doi:10.3390/molecules27196671_

Round 1

Reviewer 1 Report

The article presents the use of Cucumis Melo as biosorbent for Pb and Cu biosorption.

The article needs dome improvements:

- please clearly state the novelty of your work

- what are the advantages of your process compared to the biosorption of these heavy metals on other biosorbents

- provide a comparation to other data available on literature for Pb and Cr biosorption (q max for different biosorbents, biosorption efficiency)

Please revise:

- row 52_ "are includes"

- row 56: the best method for removal of heavy metals is biosorption which is good alternative - either the best either a good alternative

- row 62: These biosorbents are the most widely consumed beverage in the world.  Really? biosorbents are beverages?????

- row 66: The benefits of using these adsorbent is the functional groups of these organic compounds (polysaccharides, alkaloids etc) are able to interchange heavy metal ions by chemical reaction - No mention of any organic compounds previously, repetition of these

- table 1, please explain: Mean% sorption;  % sorption - how was it calculated?

- table 2: %sor Ption

- Figures 1 and 2 presents the data from Table 3. please chose one.

- there is NO discussion regarding the obtained data

Author Response

Dear worthy reviewer-1,

I hope you will be fine and find this comments response file in good health. Thanks for your suggestions to improve the manuscript. Please find attached file having reply to your comments.

Thanks and regards

Author

Reviewer 2 Report

line 78 - part 2.1. the same letters and abbreviations do not tell the potential reader. I believe that either this content should be developed, clarified or removed

picture 1 -2 -what is the description 1-3 on the OX axis of the graph, please specify it in the legend or remove the numbers

line 207-209 and relate to equations 1 and 2, 3 (line 227) I think this information should be in mwtodicy and not in the discussion of the results

Chapter 3.2 - the introduction shows that the seeds to be used for treatment are waste and the technique introduced is to reduce waste. why then the authors do not use waste but buy melons to obtain the raw material? Moreover, it may turn out that the method of grain cleaning will be so burdensome or burdensome for the environment that the good sorption of heavy metals on them will be irrelevant.

chapter 3.2 - were the seeds prepared in any way apart from the extraction, washing and drying? ground?

I would like to know the raw material cost and the cost of human labor in the degreasing of melon grains and the environmental burden. I seriously consider the efficiency and usefulness of such solutions - considering the industrial scale

There is no dissolution at work !!!!

Author Response

Dear worthy reviewer-2,

I hope you will be fine and find this comments response file in good health. Thanks for your suggestions to improve the manuscript. Please find attached file having reply to your comments.

Thanks and regards

Author

Round 2

Reviewer 1 Report

Agricultural waste as low-cost adsorbents for use in the removal of contaminants from wastewater has been previously investigated: 

https://doi.org/10.1007/s13201-021-01421-5

https://doi.org/10.1007/s002540050441 

https://doi.org/10.1016/j.biortech.2012.04.053

These could be used for comparing your results.

In terms of discussion: -what functional groups from your biosorbent is binding the heavy metals?  -what analysis have you made for showing that the metals are retained on the seeds? -analyze other process parameters that are influencing the biosorption: temperature, pH.

You have mentioned that the biosorbent is renewable and green: - how is renewable? Have you performed any regeneration studies? 

Author Response

Dear worthy reviewer,

Thanks for reviewing my manuscript to enhance its quality. The comments reply is in attached file.

Thanks and regards

Ali Irfan

Reviewer 2 Report

you can see a certain correction increasing the quality of work, but: still in chapter 2.5, when describing isothermal models, the authors provide their mathematical definition in the discussion and discussion of results instead of in mathematics Much too long conclusions, some of this content can be included in the discussion

Author Response

Respected Reviewer,

I hope you will be fine and find this comments response file in good health.  Thanks for reviewing my manuscript to convert it a quality work for scientific community.

Thanks and regards

Author
